# Exploring the Influence Factors of Early Hydration of Ultrafine Cement

**DOI:** 10.3390/ma14195677

**Published:** 2021-09-29

**Authors:** Yi Shi, Tao Wang, Haiyan Li, Shaoliang Wu

**Affiliations:** Metals & Chemistry Research Institute, China Academy of Railway Sciences Corporation Limited, Beijing 100081, China; 027wangtao@163.com (T.W.); malelihaiyan@163.com (H.L.); wushliang@sina.com (S.W.)

**Keywords:** ultrafine cement, hardening accelerators, curing condition

## Abstract

This work intends to contribute to the understanding of the influence factors of early hydration of ultrafine cement by focusing on the different fineness, different kinds of hardening accelerators, and different curing temperatures. Isothermal calorimetry, thermogravimetry, and X-ray diffraction (XRD) were performed to compare the hydration and chemical evolution of pastes containing accelerators with different fineness and curing temperatures; meanwhile, mechanical properties and water absorption were tested. The results showed that the cement fineness had a significant effect on the early hydration process; the smaller the cement particle size, the higher the early compressive strength. The 24 h compressive strength of ultrafine cement with a particle diameter of 6.8μm could reach 55.94 MPa, which was 118% higher than the reference cement. Water absorption test results indicated that adding 1% Ca(HCOO)_2_ to ultrafine cement can effectively reduce the water absorption, and it was only 1.93% at 28 d, which was 46% lower than the reference cement. An increase in curing temperature accelerated the activation of ultrafine cement in terms of the strength development rate, and the content of Ca(OH)_2_ in the ultrafine cement paste could reach 13.09% after being mixed with water for 24 h, which was 22% higher than that of the reference cement.

## 1. Introduction

Ultrafine cement particle is smaller than the ordinary ones and has better flowing performance and higher strength [1,2]; that is why it’s mainly to satisfy the demands of grouting materials and high-early strength of cement-based materials. Portland cement is widely used in tunnel engineering as a grouting material [3,4,5,6]. With the actual engineering requirements higher and higher, the weakness of Portland cement, such as large particles and poor impermeability, cannot be ignored [1,7,8,9]. Japan developed ultrafine cement as a grouting material in 1974 [10], aiming to not only fill tiny cracks but also meet environmentally friendly requirements [11]. Since then, the research on ultrafine cement has made continuous progress. It is believed that with the decrease of cement particle size, the specific surface area of cement increases, the reaction contact area increases, the hydration rate accelerates, and the strength develops rapidly [12,13,14]. According to the principle of accumulation effect, ultrafine cement has the function of filling the pores and making the cement stone more compact [15]. Under the condition of the same particle size, the particle morphology also affects the working performance of ultrafine cement [16]. To sum up, ultrafine cement work performance is related to a variety of factors. Studies have not tested the macro and micro properties of ultrafine cement systematically [17,18] and have not considered the influence of temperature and additives on the hydration process. Therefore, exploring the performance of ultrafine cement under different curing conditions and different early-strength agents can further understand the performance of ultrafine cement.

The addition of a hardening accelerator can shorten the setting time, improve the early strength [19,20], speed up mold turnover, and improve production efficiency. Thus, the research of early strength agents is of great significance for concrete prefabricated components [21,22]. Different hardening accelerators have different working principles in the cementitious material system [23].

Although the study content is very rich, unfortunately, we found no published literature concerning the performance of composites prepared using hardening accelerators with different particle sizes of ultrafine cement under different curing temperatures. Therefore, the formula of ultrafine cement needs to be optimized for better performance in application. As one high-performance material, this paper reports a new experimental study on hardening accelerator-ultrafine cement composites under different curing conditions to examine the improvement in selected engineering properties. The reaction mechanism of the three ultrafine cement with different particle sizes was also investigated.

## 2. Materials and Methods

### 2.1. Materials

The cement (C) used in this experiment is produced by China United Cement Corporation (P.I 42.5) (Beijing, China) [24]. Three kinds of fineness were selected for ultrafine cement, which were recorded as CX6000, CX7000 and CX8000. The early strength agents were analytical reagents and tap water as mix water. All experiments in this article used the same materials. The particle size distribution of cement is shown in Figure 1.

### 2.2. Methods

(1)Compressive strength: Cement, river sand, water, and additives were mixed according to ratio. Cement mortar mixer (Wuxi Jian Gong Test Equipment Co., Ltd., Wuxi, China) was used: slow mixing 120 s, fast mixing 180 s before molding. The grouted specimens (40 mm × 40 mm × 160 mm) were kept in a horizontal position in molds for 24 h at different temperatures. They were then extracted and moved to a standard curing room until testing time. The compressive strength tests were conducted on the specimens cured for 1, 28, and 90 days according to GB/T 17671–1999 [25]. Three identical samples were tested for each test, and the values were averaged.(2)Water absorption: Each group prepared three mortars (70.7 mm × 70.7 mm × 70.7 mm), cured for 28 d, then placed in the vacuum drying oven at 105 °C for 48 h. The temperature was lowered to the standard curing temperature, and then the specimens were taken out and immersed in water. It was ensured that the bottoms of mortars were fully in contact with water, with the surface below the water surface of at least 2 cm. The mortars were immersed in water for 48 h. We set the measurement time point during the period according to DL/T 5148-2012 [26] and finally obtained the water absorption curve.(3)Thermal analysis: The TAM Air eight-channel isothermal calorimeter produced by the American TA Company (New Castle, DE, USA) was used to measure the specific heat flow released during hydration at 20 °C. The samples were temperature-equilibrated for 72 h prior to the measurements. Mixing was performed for 1 min at 860 rpm. Afterwards, approximately 1.8 g of paste was cast into ampoules and inserted into the calorimeter [27].(4)Thermogravimetric analysis: TG/DTG was done by used METTLER TOLEDO TGA 2 thermal analyzer (METTLER TOLEDO, Zurich, Switzerland). Dried cement paste was taken in a ceramic crucible and heated from room temperature to 1050 °C at a heating rate of 10 °C/min under a nitrogen flow of 70 mL/min [28,29,30,31,32,33,34].(5)X-ray diffraction: XRD was conducted using PANalytical X’Pert PRO (PANalytical, Almelo, Netherlands) using a Cu Kα radiation nickel (PANalytical, Almelo, Netherlands) foil filter. Dried cement paste was placed on a flat plate sample geometry. A fixed divergence slit of 1/16 degree was chosen to limit beam overflow on the samples at small angles of 2θ. The X’Celerator high-speed linear detector was used for the XRD experiment, and scans ranged from 5 to 70 degrees [27,35].

## 3. Results and Discussions

### 3.1. Compressive Strength

The w/c ratio of the mortar samples was 0.3, and the mortar expansion was 120–150 mm. Na_3_SiO_3_, KAl(SO_4_)_2_, and Ca(HCOO)_2_ (Na, K and Ca)were selected as hardening accelerators. River sand (S) with maximum particle size of ≤4.75 mm and fineness modulus of 2.6~2.8 were selected to prepare mortar samples. In the early stage of the test, the three hardening accelerators were introduced at 1% and 2%, and the 24 h compressive strength was tested. The results show that 1% dosage was the best, so the dosage of hardening accelerators in the subsequent test groups was 1%. The mortar mix ratio is shown in Table 1. The ionic concentration of the tap water is shown in Table 2.

In order to further explore the factors affecting the early strength of ultrafine cement, a 60 °C steam curing condition was added for the Ca-1% experimental group. The compressive strength of the specimens of different ages of each experimental group is shown in Figure 2.

It can be seen from Figure 2 that in this test, ultrafine cement can reach the strength of reference cement (P.I 42.5) under steam curing conditions in 24 h without hardening accelerator or 60 °C curing temperature, and the later strength can also reach a high level. The steam curing caused the strength shrinkage of ultrafine cement in the later stage. In conclusion, ultrafine cement had good strength performance under standard curing conditions.

### 3.2. Water Absorption

In order to explore the influence of hardening accelerators and curing system on the water absorption of ultrafine cement, three kinds of hardening accelerators were added into three kinds of ultrafine cement with 1% dosage, and the reference cement was set as reference. The mix proportion is shown in Table 1.

The experiment explored the influence of fineness, hardening accelerators, and curing system on the capillary water absorption of ultrafine cement. The results are shown in Figure 3. The test times were set to 0 min, 10 min, 30 min, 1 h, 10 h, 24 h, and 48 h.

It can be seen from Figure 3 that the water absorption rate of ultrafine cement was low. Adding 1% Ca(HCOO)_2_ can effectively reduce water absorption and improve durability. The total water absorption of mortar specimens considers two major factors; one is the water absorption performance of the constituent materials, and the other is the surface porosity and pore characteristics of the specimen. Ultrafine cement particles are small and require a large amount of water for hydration reaction. At the same water–binder ratio, the free water in the system is lower than that of the reference cement, so there are fewer pores left during the evaporation process, which reduces the water absorption rate. On the other hand, when Ca(HCOO)_2_ was added to the ultrafine cement system, the flowability was significantly improved, which further compensated for the fluidity problem of the ultrafine cement, and the final total water absorption rate decreased.

### 3.3. Thermal Analysis

In this experiment, only three ultrafine cement blank groups were selected for experimental testing at 20 °C. The w/b of cement slurry was 0.3. We took an amount of cement slurry with a dropper into an ampoule, put it into an isothermal calorimeter, and observed the hydration exotherm. The 72 h heat of hydration test result is shown in Figure 4.

According to Figure 4, the cumulative heat release of ultrafine cement was always higher than that of the reference cement in the same time period, but there were nearly no differences among the ultrafine cements. All the experimental groups experienced four hydration exothermic stages at the initial stage of hydration, namely “Accelerated hydration–Latent period–Reaccelerated hydration–Slow hydration period”. In the very early stage of hydration acceleration, the rate was very fast, and the time was too short to be seen in the graph; in the 0–5 h time period, the hydration rate values of the four experimental groups were all low; after 5 h, all the experimental groups entered the “reacceleration” stage, and the ultrafine cement had a significantly higher heat release than the reference cement during this stage; after 20 h, each group entered a flat period, the hydration rate decreased, and the hydration rate curve and the cumulative heat of hydration curve tended to be flat. To sum up, the ultrafine cement particles had a large specific surface area, which made it easier to react with water and release more heat. In the period of 0–72 h of early hydration, the hydration process of ultrafine cement was fast, and the hydration degree was high. It can be considered that the degree of hydration of ultrafine cement is higher than reference cement, which is the reason for the rapid development of its early strength.

### 3.4. Thermogravimetric Analysis

In this section, the thermogravimetric analysis method was used to discuss the influence of different hardening accelerators and different particle sizes on the hydration process of the cementitious system at different curing temperatures through the content of CH and chemically bound water in the paste specimens. The proportion of each experimental group is shown in Table 3, and TG and DTG curves of each age are shown in Figure 5.

By comparing the TGA curves of three ages, it was found that with the increase of age, the influence of different hardening accelerators on ultrafine cement became smaller and smaller. The early TG curves showed that the group added with Na_3_SiO_3_ had the least amount of mass loss and the lowest degree of hydration, but it was found that in the latter, there was no significant difference among the samples with different types of hardening accelerators. Under 60 °C curing temperature, the early hydration degree of each group was significantly improved, and the decomposition peak of AFm appeared earlier than the standard maintenance group at the same period. However, as the age increased, the differences caused by different maintenance systems gradually decreased. Under the same curing conditions, the hydration degree of reference cement was lower than that of ultrafine cement. In order to further explore the changes of CH and chemical-bonded water content in each age of ultrafine cement series, the hydration degree of each experimental group was calculated according to the thermogravimetric test results, which is shown in Figure 6.

It can be seen from Figure 6 that the early hydration degree of ultrafine cement was higher than that of the reference cement in the same period. With the increase of age, the hydration degree of the reference cement increased rapidly and reached the same level as the ultrafine cement at 28 days. Under standard curing conditions, the addition of hardening accelerators did not significantly promote the hydration of ultrafine cement. The results of the thermogravimetric analysis show that the early hydration level of ultrafine cement can be high without adding any hardening accelerator under standard curing conditions, and the later hydration level continues to improve.

### 3.5. X-ray Diffraction

X-ray diffraction was used to analyze the types and changes of hydration products in different experimental groups at different ages and to further explore the factors influencing the hydration process of ultrafine cement. The results are shown in Figure 7.

The results show that under the same curing conditions, the phase types of each experimental group were the same at the same age. Small amounts of CH and Ettringite (Ett) were formed in CX8000 cement at 1 d, and the peak of CH was obvious at 28 d. With the decrease of particle size of ultrafine cement, the clinker content was lower and lower, and more and more CH was generated. At the same ratio, the peak value of each cement clinker in the experimental group under steam curing conditions was lower. In summary, as the particle size decreases, the degree of hydration of the ultrafine cement increases at the same age, but it does not affect the types of phases in the slurry. Adding an early strength agent can accelerate the hydration rate of some clinkers. Steam curing conditions can promote the hydration of cement clinker.

## 4. Conclusions

The current study investigated the effects of different fineness, different early strength agents, and different curing conditions on the early hydration of ultrafine cement. Based on experimental results, the following conclusions can be made:Under the standard curing condition, ultrafine cement can reach high strength in 24 h without adding an early strength agent, and the CX8000 ultrafine cement can reach 55.94 MPa, which is 118% higher than the reference cement. Adding 1% Ca(HCOO)_2_ can effectively reduce the water absorption rate of ultrafine cement.The hydration heat test results showed that with the decrease of cement particle size, the early hydration speed was accelerated, and the hydration process was significantly accelerated compared with the reference cement. The maximum hydration heat release of ultrafine cement in 72 h was 262.55 J/g, which was 12% higher than that of reference cement.The results of thermogravimetric analysis and XRD analysis showed that the hydration products of ultrafine cement paste were the same as those of the reference cement. Under steam curing conditions, the CH content of ultrafine cement could reach 13.09% in 1 day, which was 22% higher than that of reference cement and 33% higher than that of ultrafine cement under standard curing conditions.

## Figures and Tables

**Figure 1 materials-14-05677-f001:**
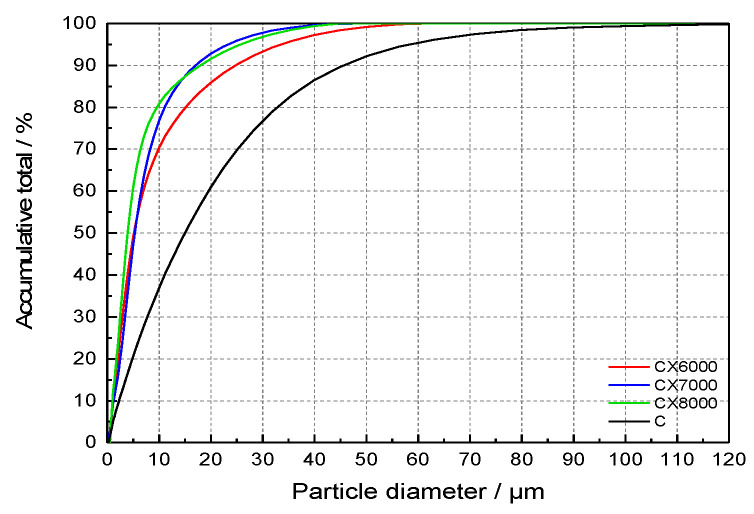
Cement particle size distribution.

**Figure 2 materials-14-05677-f002:**
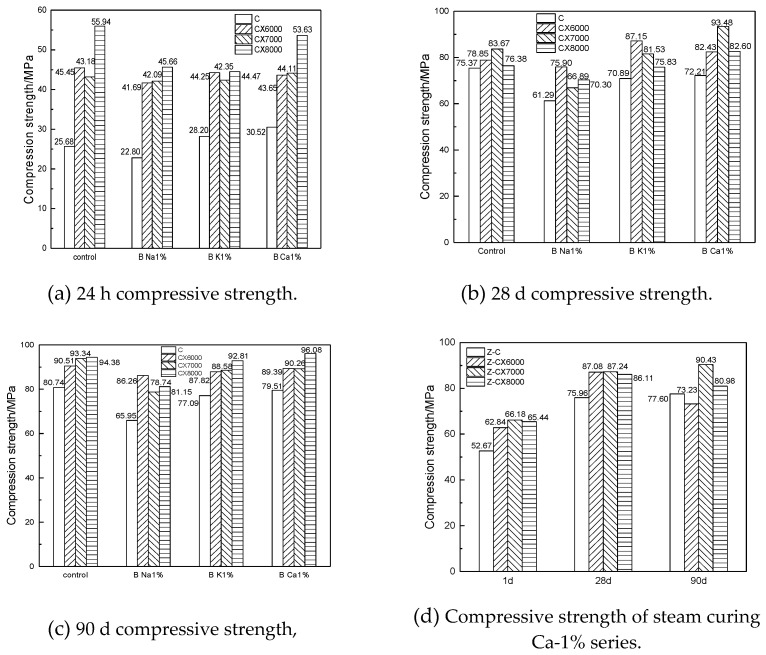
Ultrafine cement mortar compression strength with admixture/temperature at different ages.

**Figure 3 materials-14-05677-f003:**
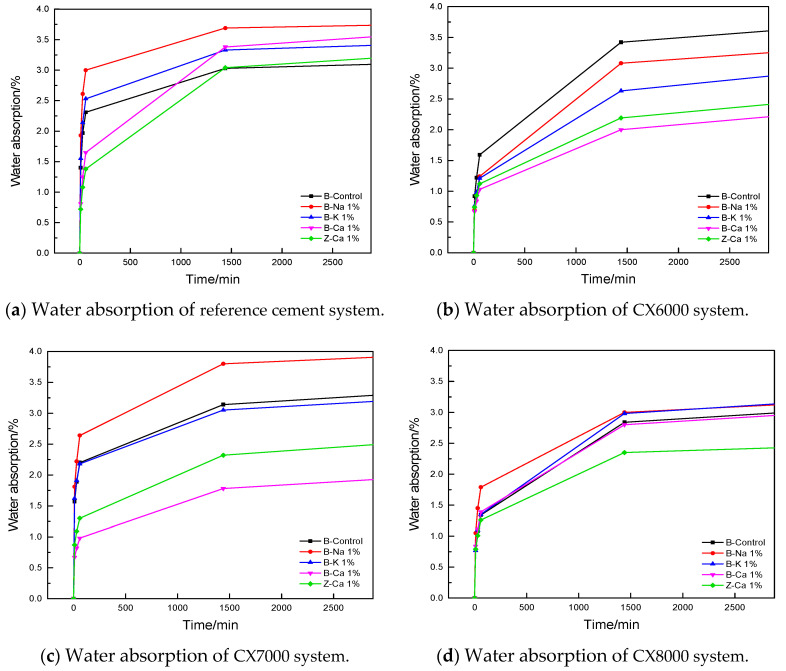
Water absorption of ultrafine cement.

**Figure 4 materials-14-05677-f004:**
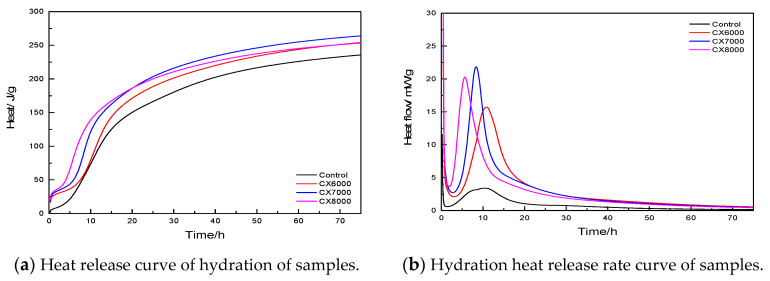
Heat of hydration curves of ultrafine cement.

**Figure 5 materials-14-05677-f005:**
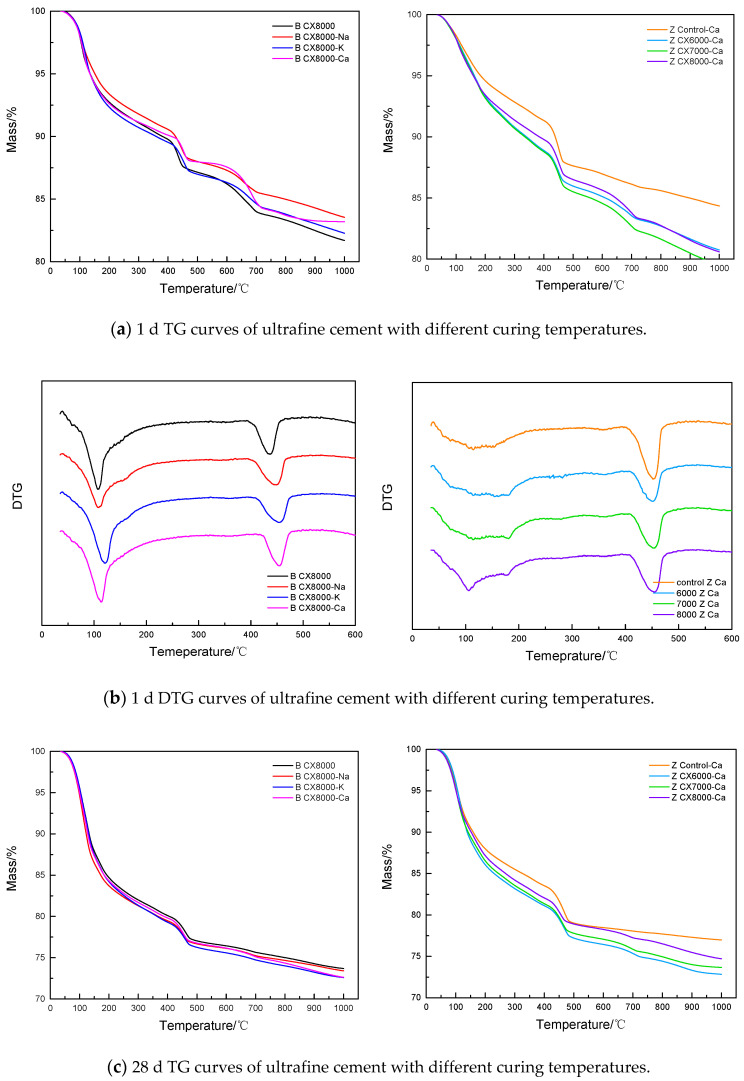
TG/DTG curves of ultrafine cement system at different ages.

**Figure 6 materials-14-05677-f006:**
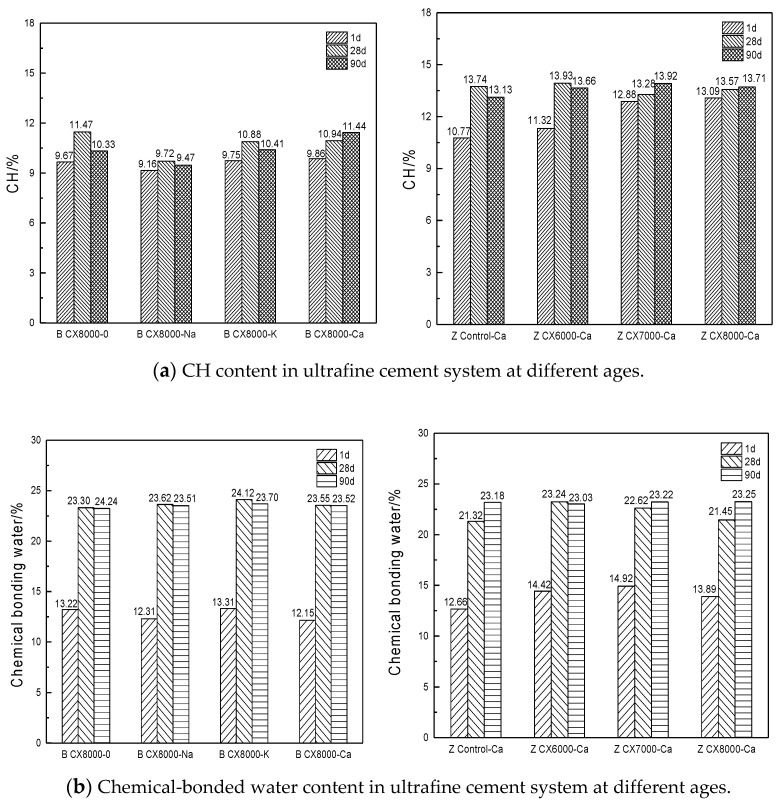
CH and chemical-bonded water content in ultrafine cement system.

**Figure 7 materials-14-05677-f007:**
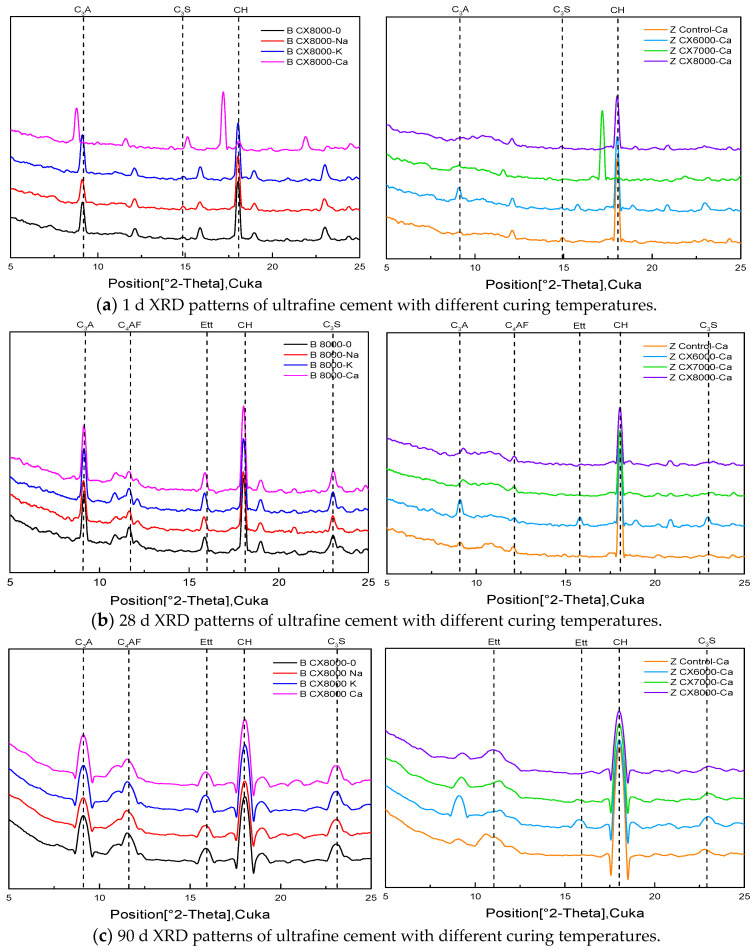
XRD analysis of ultrafine cement system at different ages.

**Table 1 materials-14-05677-t001:** Ultrafine cement (CX6000) with hardening accelerators mortar mix ratio.

Sample	CX6000	W/B	S/B	Na_2_SiO_3_	KAl(SO_4_)_2_	Ca(HCOO)_2_
0. Control	100	0.3	1.64	--	--	--
1.Na-1%	100	0.3	1.64	1.0	--	--
2.K-1%	100	0.3	1.64	--	1.0	--
3.Ca-1%	100	0.3	1.64	--	--	1.0

**Table 2 materials-14-05677-t002:** Ionic concentration of the tap water.

	Ca^2+^	Mg^2+^	Na^+^	K^+^	Al^3+^	Fe^3+^/Fe^2+^	Cl^−^	SO_4_^2−^
Concentration mg/L	2.1	0.9	4.0	0.5	<0.005	<0.05	0.4	26.7

**Table 3 materials-14-05677-t003:** Ultrafine cement with different accelerators and particle sizes system mortar mix proportion.

	Cement	Na_3_SiO_3_	KAl(SO_4_)_2_	Ca(HCOO)_2_	W/B
0.CX8000	100	——	——	——	0.3
1.CX8000 Na	100	1.0	——	——	0.3
2.CX8000 K	100	——	1.0	——	0.3
3.CX8000 Ca	100	——	——	1.0	0.3
4.C Ca	100	——	——	1.0	0.3
5.CX6000 Ca	100	——	——	1.0	0.3
6.CX7000 Ca	100	——	——	1.0	0.3

## Data Availability

Data is contained within the article.

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
