# Peer review of "Exploring the Influence Factors of Early Hydration of Ultrafine Cement"

_materials, 2021, doi:10.3390/ma14195677_

Round 1
Reviewer 1 Report
Dear authors
The topic of this research is interesting, however tehere are some issues with your paper that should be adressed before its acceptance. I think you omit some important references in the introduction, being ultrafine cements a promising material, why no one has tested basic properties before.
Besides, some point that is unaceptable is the use of tap water in the cement mixture. Tap water change composition from one city to another. Using tap water makes, for example, not possible to replicate accurately your research, which is mandatory for any scientific paper. This is a serious issue.

Author Response
Response to Reviewer 1 Comments
Thanks for your suggestions. We have made the adjustment as your request, please check and let me know if there is anything else needed to adjust.
Reviewer 1 comments:
I think you omit some important references in the introduction, being ultrafine cements a promising material, why no one has tested basic properties before.
Response 1:
Existing studies have basically focused on the effect of a single influencing factor on the properties of ultrafine cement, such as particle fineness, hardening accelerators, curing temperature, etc. There is a lack of research on the comprehensive impact of the above factors on ultrafine cement.
Reviewer 2 comments:
Tap water change composition from one city to another. Using tap water makes, for example, not possible to replicate accurately your research, which is mandatory for any scientific paper. This is a serious issue.
Response 2:
In consideration of the fact that most of the construction sites use tap water as mixing water,tap water is selected in this paper. Please allow me to supplement the explanation to you with the ionic concentration of the tap water in this paper.
Table 1. Ionic concentration of the tap water
|
|
Al3+ |
Fe3+/Fe2+ |
Zn2+ |
Cl- |
SO42- |
|
Concentration mg/L |
<0.005 |
<0.05 |
<0.05 |
0.4 |
26.7 |
We appreciate the careful reading of our manuscript and valuable suggestions of the reviewer. Hope these will make it more acceptable for publication. Please see the attachment.

Reviewer 2 Report
This manuscript could be a good paper if the section related to Methods is significantly improved.
Line 25: Please, describe cement as Portland cement.
Lines 26-28: Please, add a reference.
Lines 56-57: Please, add a reference.
Figure 1 is fuzzy. Please, improve the quality of Figure 1.
1.1 Methods must be significantly improved. The authors need to re-write this section elaborately. The description of this section is too simple to explain Test Method.
For compressive strength, water absorption, thermal analysis, thermogravimetric analysis, which standard methods did the authors employ?
Line 75: Please, add a reference about American TA Company and a detailed description of what the authors did with it.
Line 82: Please, add the description why Cu Ka radiation source used? Is there any references associated with it?
Author Response
Response to Reviewer 1 Comments
Thanks for your suggestions. We have made the adjustment as your request, please check and let me know if there is anything else needed to adjust.
Reviewer 1 comments:
Line 25: Please, describe cement as Portland cement.
Lines 26-28: Please, add a reference.
Lines 56-57: Please, add a reference.
Figure 1 is fuzzy. Please, improve the quality of Figure 1.
Response 1:
We have received the comments on our manuscript. According to the comments of the reviewers, we have revised our manuscript. The revised manuscript and the detailed responses to the comments of these reviewers are attached.
Reviewer 2 comments:
1.1 Methods must be significantly improved. The authors need to re-write this section elaborately. The description of this section is too simple to explain Test Method.
For compressive strength, water absorption, thermal analysis, thermogravimetric analysis, which standard methods did the authors employ?
Line 75: Please, add a reference about American TA Company and a detailed description of what the authors did with it.
Line 82: Please, add the description why Cu Ka radiation source used? Is there any references associated with it?
Response 2:
(1) Compressive strength: Cement, river sand, water and additives were mixed according to ratio. Used cement mortar mixer, slow mixing 120 s, fast mixing 180 s before molding. The grouted specimens (40mm × 40mm × 160mm) were kept in a horizontal position in molds for 24 hours at different temperatures. They were then extracted from them and moved them in standard curing room until testing time. The compressive strength tests were conducted on the specimens cured for 1, 28, and 90 days according to GB/T 17671-1999[26]. Three identical samples were tested for each test and the values were average.
(2) Water absorption: Each group prepared three specimens (70.7mm × 70.7mm × 70.7mm) , cured for 28d, then placed in the vacuum drying oven at 105 ℃ for 48h then lower the temperature to the standard curing temperature. Taken the specimens out and immersed in water. Ensure that the bottom of mortars were fully in contact with water, the surface below the water surface of at least 2cm. The mortars were immersed in water for 48h. Set the measurement time point during the period according to DL/T 5148-2012[27], and finally obtained the water absorption curve.
(3) Thermal analysis: The TAM Air eight-channel isothermal calorimeter produced by American TA Company was used to measure the specific heat flow released during hydration at 20℃. The samples were temperature-equilibrated for 72 h prior to the measurements. Mixing was performed for 1 min at 860 rpm. Afterwards approximately 1.8 g of paste was cast into ampoules and inserted into the calorimeter[28].
(4) Thermogravimetric analysis: TG/DTG was done by used METTLER TOLEDO TGA 2 thermal analyzer. Dried cement paste was taken in a ceramic crucible and heated from room temperature to 1050℃ at a heating rate of 10℃/min under a nitrogen flow of 70 mL/min[29-35].
(5) X-ray diffraction: XRD was conducted using PANalytical X’Pert PRO using a Cu Kα radiation nickel foil filter. Dried cement paste was placed on a flat plate sample geometry. Fixed divergence slit of 1/16 degree was chosen to limit beam overflow on the samples at small angles of 2θ. The X’Celerator high-speed linear detector was used for XRD experiment and scan range from 5 to 70 degrees[28,36].
We appreciate the careful reading of our manuscript and valuable suggestions of the reviewer. Hope these will make it more acceptable for publication. Please see the attachment.

Reviewer 3 Report
Dear Authors, congratulations for your work which I found particularly interesting from a clinical point of view.
I just have some minor suggestions to improve the quality of your work:
- I think that the introduction should be further expanded, if possible
- At the end of the introduction you should report the statistical null hypothesis of the study
- Captions of figures and tables should be further expanded
- the conclusions are too long. I think that a small paragraph of few lines not divided into different points is much better for the reader. If you prefer to leave the three points separated you can do that but I suggest to reduce very much the description!
- I think that 16 references are not enough for this kind of articles. Please improve your introduction and/or discussion taking into account all the relevant literature.
Kind regards,
The Reviewer
Author Response
Response to Reviewer 1 Comments
Thanks for your suggestions. We have made the adjustment as your request, please check and let me know if there is anything else needed to adjust.
Reviewer 1 comments:
I think that the introduction should be further expanded, if possible
At the end of the introduction you should report the statistical null hypothesis of the study
Captions of figures and tables should be further expanded
Response 1:
We have received the comments on our manuscript. According to the comments of the reviewers, we have revised our manuscript. The revised manuscript and the detailed responses to the comments of these reviewers are attached.
Reviewer 2 comments:
the conclusions are too long.
Response 2:
- Under the standard curing condition, ultrafine cement can reach high strength in 24 hours without adding early strength agent, and the CX8000 ultrafine cement can reach 55.94MPa, 118% higher than the reference cement. Adding 1% Ca(HCOO)2 can effectively reduce the water absorption rate of ultrafine cement.
- The hydration heat test results showed that with the decrease of cement particle size, the early hydration speed was accelerated, and the hydration process was significantly accelerated compared with the reference cement. The maximum hydration heat release of ultrafine cement in 72h was 262.55 J/g, which was 12% higher than that of reference cement.
- The results of thermogravimetric analysis and XRD analysis showed that the hydration products of ultrafine cement paste were the same as those of the reference cement. Under steam curing condition, the CH content of ultrafine cement can reach 13.09% in 1 day, which was 22% higher than that of reference cement and 33% higher than that of ultrafine cement under standard curing condition.
Reviewer 3 comments:
I think that 16 references are not enough for this kind of articles. Please improve your introduction and/or discussion taking into account all the relevant literature.
Response 3:
We have improved our introduction and methods. The revised manuscript and the detailed responses to the comments of the one reviewer are attached.
We appreciate the careful reading of our manuscript and valuable suggestions of the reviewer. Hope these will make it more acceptable for publication. Please see the attachment.

Round 2
Reviewer 1 Report
Dear authors
Please include the table with the ionic concetration of the tap water in the text. Aslo it is advisable that you include other elements that are known to have effect on cement: Ca, Mg, Na, K...
Author Response
Dear reviewer,
We greatly appreciate your suggestion to this paper. We hope that the revised manuscript is now suitable for publication.
Reviewer 1 comments:
Please include the table with the ionic concetration of the tap water in the text. Aslo it is advisable that you include other elements that are known to have effect on cement: Ca, Mg, Na, K...
Response 1:
Table 1. Ionic concentration of the tap water
|
|
Ca2+ |
Mg2+ |
Na+ |
K+ |
Al3+ |
Fe3+/Fe2+ |
Cl- |
SO42- |
|
Concentration mg/L |
2.1 |
0.9 |
4.0 |
0.5 |
<0.005 |
<0.05 |
0.4 |
26.7 |
Thank you again for your valuable comments and suggestions. Please see the attachment.
Sincerely yours,
The Authors

Reviewer 2 Report
Thank you for the authors' hard work.
This manuscript is significantly improved.
For me, this manuscript can be published in Materials.
Author Response
Dear reviewer,
Thank you for your careful review. We really appreciate your efforts in reviewing our manuscript during this challenging time. We wish good health to you. Your careful review has helped to make our study clearer and more comprehensive.
Sincerely yours,
The Authors